# The Profile and Clinical Significance of ITGB2 Expression in Non-Small-Cell Lung Cancer

**DOI:** 10.3390/jcm11216421

**Published:** 2022-10-29

**Authors:** Lingling Zu, Jinling He, Ning Zhou, Jingtong Zeng, Yifang Zhu, Quanying Tang, Xin Jin, Lei Zhang, Song Xu

**Affiliations:** 1School of Life Sciences, Tianjin University, Tianjin 300072, China; 2Tianjin Key Laboratory of Lung Cancer Metastasis and Tumor Microenvironment, Tianjin Lung Cancer Institute, Tianjin Medical University General Hospital, Tianjin 300052, China; 3Department of Lung Cancer Surgery, Tianjin Medical University General Hospital, Tianjin 300052, China

**Keywords:** non-small-cell lung cancer, integrins, ITGB2, epithelial–mesenchymal transition (EMT), metastasis

## Abstract

Integrins are involved in extracellular and intracellular signaling and are often aberrantly expressed in tumors. Integrin beta 2 (ITGB2) has previously been demonstrated to be correlated with the host defense. However, the expression profile and role of ITGB2 in non-small-cell lung cancer (NSCLC) remain unclear. Here, we found that the genetic alterations in ITGB2 was predominated by gene mutation and copy number deletion using cBioPortal analysis, and its expression was downregulated in the NSCLC tissues, as validated by the UALCAN, TCGA, and GEO databases and our tissue samples. Kaplan–Meier (KM) plotter analysis revealed that patients with a lower ITGB2 expression had a shorter overall survival (OS) time (*p* = 0.01). Moreover, 1089 differentially expressed genes (DEGs) in the NSCLC tissues were screened using the TCGA database. The GO and KEGG enrichment analysis showed that the DEGs were closely associated with immune processes and cell adhesion. The protein–protein interaction (PPI) network revealed that 10 of 15 EMT-related genes among the DEGs might lead to the metastasis of NSCLC. Concomitantly, the expression of ITGB2 was positively correlated with the infiltration of Treg cells and Myeloid-derived suppressor cells (MDSC). Biologically, the ectopic expression of ITGB2 significantly inhibited the proliferation and metastasis of NSCLC cells. Mechanistically, we demonstrated that ITGB2 suppressed the expression of N-cadherin, Vimentin, Slug, Snail, and Twist, while it promoted E-cadherin expression, according to gain-of-function studies. In conclusion, ITGB2 can inhibit the proliferation and migration of NSCLC cells, leading to a poor prognosis, via epithelial–mesenchymal transition (EMT) signaling.

## 1. Introduction

Lung cancer is one of the most prevalent tumors and contributes to the largest number of cancer-related deaths worldwide [1]. Non-small-cell lung cancer (NSCLC) is the most common type of lung cancer and accounts for a proportion of nearly 85% of all patients, and it takes two main forms: lung adenocarcinoma (LUAD) and lung squamous cell carcinoma (LUSC) [2,3]. With the development of therapeutic strategies, emerging treatments, such as targeted therapy and immunotherapy, have successfully improved the prognosis of patients with NSCLC [4,5,6,7]. However, there are still many patients who cannot benefit from those treatments [8]. Therefore, the exploration of new targets and biomarkers remains urgent in the case of NSCLC.

It has been proved that the tumor microenvironment (TME) plays an important role in cancer progression and influences the therapeutic outcome [9]. TME consists of multiple cell types, extracellular matrices, and other molecules. These cell types include tumor cells and non-tumorous cells, such as immune cells and non-immune cells [10]. The importance of immune and non-immune cells in regulating tumor cell proliferation, influencing tumor cell differentiation, and controlling the death of tumor cells has been widely recognized [11]. In addition, previous studies have also validated the therapeutic effect of treatments based on targeting TME [12,13,14].

Integrins are a cell surface glycoprotein family, which interact with the extracellular matrix and ignite a series of cellular responses, such as cell proliferation, differentiation, and migration [15]. Integrin beta 2 (ITGB2), also named as CD18, is one of the integrin subunits and was first observed on the surface of leukocytes. Several studies have suggested that ITGB2 possesses the function of promoting leukocyte adhesion and extravasation [16,17]. It was also observed that ITGB2 indirectly promotes the proliferation of oral squamous cell carcinoma by regulating cancer-associated fibroblasts [18] and mediates YAP-induced breast cancer cell transendothelial invasion [19]. In addition, researchers found that ITGB2 is related to the prognosis of gliomas and is a potential biomarker for immunotherapy [20]. However, the expression profile and the role of ITGB2 in NSCLC remain unknown.

In the present study, we found that the expression level of ITGB2 in NSCLC tumor tissues was lower than that in normal tissues, and its low expression was related to an inferior prognosis. We also confirmed that ITGB2 regulates tumor cell proliferation, invasion, and migration through experiments in vitro. In addition, we found that the expression of ITGB2 was associated with the infiltration level of immune cells. In conclusion, ITGB2 is a new target for NSCLC treatment and a potential biomarker for immunotherapy.

## 2. Materials and Methods

### 2.1. Data Collection and Processing

The gene expression and RNA sequencing data of NSCLC were assessed from the TCGA (https://portal.gdc.cancer.gov/ (accessed on 1 May 2022) and Gene Expression Omnibus (GEO) databases (https://www.ncbi.nlm.nih.gov/geo/ (accessed on 1 May 2022). In total, 4 cohorts were collected, including TCGA-LUAD and LUSC, GSE120622, GSE116959, and GSE19188, composed of early stage and advanced-stage patients. The mRNA expression profiles of 99 lung cancer cell lines in the Cancer Cell Line Encyclopedia (CCLE) (https://sites.broadinstitute.org/ccle (accessed on 1 May 2022) were downloaded for the analysis. The cBio Cancer Genomics Portal (c-BioPortal) (http://cbioportal.org (accessed on 1 May 2022) includes a multidimensional cancer genomics dataset [21]. Therefore, the c-BioPortal database was applied to explore ITGB2 genomic alterations in LUAD and LUSC.

### 2.2. UALCAN Analysis

The UALCAN website [22] provides an extensive and interactive study of bioinformatics based on RNA-seq and clinical data of 31 malignancies from the TCGA (http://ualcan.path.uab.edu/ (accessed on 1 May 2022). The database can be used to compare the gene expressions of tumors with normal tissues, as well as different tumor stages, subtypes, and other clinicopathological features. This research, respectively, We explored examines the ITGB2 expression levels based on clinical features, such as the tissue type (normal/tumor) and clinical stage (stages I, II, III, and Ⅳ).

### 2.3. Kaplan–Meier Plotter (Lung Cancer)

The KM plotter [23] (http://kmplot.com/analysis/ (accessed on 1 May 2022) was able to evaluate the survival prognosis of the related genes by mapping the survival curve using 1925 NSCLC samples. The prognostic significance of ITGB2 in NSCLC, including the overall survival (OS), first progression (FP), and post-progression survival (PPS), were investigated using this database. The hazard ratio (HR), with 95% confidence intervals, was also estimated, as well as the log-rank *p*-value. Statistical significance was defined as *p* < 0.05.

### 2.4. Quantitative Reverse-Transcription Polymerase Chain Reaction

We extracted total RNA from cell lines with the TRIzol reagent (Invitrogen), and used Prime Script RT Master Mix (TaKaRa) to synthesized cDNA. Then, reverse transcription-polymerase chain reaction (PCR) was performed using TB Green Premix ExTaq (TaKaRa) on the 7900HT Fast Real-Time PCR System (Applied Biosystems, Foster City, CA, USA; Thermo Fisher Scientific). QRT-PCR was performed to detect mRNA levels of ITGB2, and GAPDH. GAPDH was used as the endogenous control. The ITGB2 primer sequence is as follows: forward 5′-CTCTCTCAGGAGTGCACGAA-3′ and reverse 5′-CCCTGTGAAGTTCAGCTTCTG-3′. GAPDH: forward 5′-GGAGCGAGATCCCTCCAAAAT-3′ and reverse 5′-GGCTGTTGTCATACTTCTCATGG-3′. Each gene was repeated three times.

### 2.5. Western Blotting

We collected cell lines and lysed them in RIPA lysis buffer with PMSF (Beyotime, Shanghai, China). Additionally, we separated the protein samples by sodium dodecyl sulfate–polyacrylamide gel electrophoresis and transferred them to the polyvinylidene difluoride membranes (Millipore). The membranes were blocked for 2 h with 5% BSA. Subsequently, they were incubated with primary antibodies at 4 °C overnight and then incubated with the corresponding secondary antibody. The bands were visualized using enhanced chemiluminescence reagents (Yeasen, Shanghai, China). Western blotting was performed to detect the protein expression levels of ITGB2, E-cadherin, N-cadherin, Vimentin, Slug, Snail, Twist, and GAPDH. GAPDH was used as the endogenous control.

### 2.6. Immunohistochemistry

The sections of 10 pairs of LUADs, with their matched normal lung tissues, were placed in an oven at 56 °C for 1 h to melt the paraffin and prevent the tissues from shedding before dewaxing. Then, we removed the paraffin with xylene and alcohol and placed the sections in a sodium citrate buffer (pH 6.0) for repair. We placed the sections in a pressure cooker for 15 min and then cooled them to room temperature, and then washed them with phosphate-buffered saline (PBS) 3 times. We placed the sections in 3% hydrogen peroxide solution and incubated them for 15 min at room temperature to deactivate the endogenous peroxidase. The sections were incubated with anti-ITGB2 primary antibody, which was diluted in the ratio of 1:25, at 4 °C overnight. Next, the sections were incubated with secondary antibody for 1 h and then were washed 3 times with PBS. Diaminobenzidine tetrachloride (DAB) was added and then removed after we observed a satisfactory color development. We then stained the sections with hematoxylin for 25 s and washed them for 20 min. Finally, we covered the sections with neutral balsam and coverslips. Additionally, we consulted two pathologists to ensure the typicality of the selected tissues.

### 2.7. Clinical Samples and Cell Culture

Forty paired frozen fresh tumor tissues and adjacent non-tumor lung tissues were collected from patients with lung adenocarcinoma at Tianjin Medical University General Hospital (Tianjin, China). The fresh tissues were snap-frozen in liquid nitrogen. All the samples were collected with informed consent from the patients, and all the related procedures were performed with the approval of the internal review and ethics committee of the Tianjin Medical University General Hospital (ethical No. IRB2022-WZ-026). The lung cancer cell lines (H1792, H1975, PC9, H1299, A549, DV90, and SK-MES-1) and the human normal lung epithelial cell line (HBEC) were obtained from the American Type Culture Collection (ATCC). The HBEC cells were cultured in 1640 medium with 10% fetal bovine serum (FBS, Gibco, Grand Island, NY, USA) and 1% ampicillin and streptomycin. The lung cancer cell lines were cultured in DMEM medium or 1640 medium (Gibco, Grand Island, NY, USA) supplemented with 10% fetal bovine serum, 1% ampicillin, and streptomycin at 37 °C and under 5% CO_2_.

### 2.8. Analysis of Differentially Expressed Genes

The differential gene expression analysis was conducted based on the TCGA dataset using the R package DESeq2 [24]. RNA sequencing data for the normal tissue and tumor tissues were used for the analyses. In particular, the DE analysis was performed by comparing the gene expression between the tumors and normal tissues. A volcano plot was drawn to show the fold change and *p*-values of the DE genes between the two comparison groups. Significantly upregulated and downregulated genes were defined as those with an adjusted *p*-value of <0.05 and a fold change greater than 2, i.e., log2 (fold change) >1 or <−1 for up- and downregulated genes, respectively. A Venn diagram was then plotted to illustrate the subset of DEGs and epithelial mesenchymal transition (EMT). Of these overlapping genes, only the protein-coding genes were further analyzed. To elucidate the enrichment of the genes in terms of the biological processes and signaling pathways, in silico analyses were carried out, including hierarchical clustering with a heatmap, a gene ontology (GO) analysis, and Kyoto Encyclopedia of Genes and Genomes (KEGG) analysis [25].

### 2.9. Integrated Network and Gene Set Enrichment Analysis

We used ComPPI, the compartmentalized protein–protein interaction database (https://comppi.linkgroup.hu/downloads (accessed on 1 May 2022), to explore the integrated network of ITGB2. Additionally, we performed a pathway analysis using the Gene Set Enrichment Analysis (GSEA) software (version 4.3.0) (http://www.broadinstitute.org/gsea (accessed on 1 May 2022). The enriched gene sets were selected based on a false discovery rate (FDR) of <0.05 and a family-wise error rate of <0.05.

### 2.10. Cell Proliferation Analysis

For the CCK-8 assay, the cells were seeded on 96-well plates at the density of 3000 cells/well for the indicated lengths of time. The cell viability was detected using the Cell Counting Kit-8 (CCK8) by measuring the absorbance at 450 nm with a microplate reader.

For the EdU assay, the cells were seeded on 12-well plates for 48 h. Then, the cells were incubated with EdU reagent for 2 h, fixed with 4% paraformaldehyde and 0.5% Triton X-100, and finally stained by Hoechst. The EdU incorporation rate was defined as the proportion of EdU-positive cells (Red) to total Hoechst33342-positive cells (Blue). The experiment was repeated three times to conduct the calculation.

### 2.11. Transwell Invasion Assay

We performed the cell invasion assay using transwell chambers, which were pre-treated with matrix gel (Millipore). The cells were collected and seeded in the upper chamber at the density of 1.2 × 10^5^ cells per well with serum-free medium, while the medium in the bottom chamber contained 10% FBS. The chambers with invaded cells were fixed with 4% formaldehyde for 30 min, and the cells were stained with 0.1% crystal violet for 30 min, while the noninvaded cells were wiped with cotton swabs. The experiment was repeated three times to conduct the calculation.

### 2.12. Statistical Analysis

All data were shown as the mean values ± S.D. of three independent experiments. The statistical analysis was conducted using GraphPad Prism 9.0 software (La Jolla, CA, USA). Significant differences between groups were evaluated using the Student’s *t*-test, and *p* < 0.05 was considered to be statistically significant.

## 3. Results

### 3.1. ITGB2 Expression Was Downregulated in NSCLC and Associated with OS in the Public Databases

Gene mutations and CNA can alter the gene expression profiles of cancer cells, being closely associated with tumor development and progression. Thus, we firstly investigated the frequency and category of the ITGB2 gene alterations in NSCLC using the cBioPortal analysis. The results showed that the genetic alterations in ITGB2 are predominated by the mutation and deep deletion of both LUAD and LUSC in the TCGA cohorts and, significantly, more than 2% of ITGB2 alterations (including “deep deletion” and “mutation”) were detected in the LUAD and LUSC patients, as shown in Figure 1A. Additionally, we observed similar results in OncoSG and CPTAC datasets (Figure 1B). The results strongly implied the downregulation of ITGB2 in the development of NSCLC. Then, to further explore the expression profile of ITGB2 in NSCLC, we collected data from the TCGA and GEO databases. The analysis of the TCGA cohort showed that ITGB2’s mRNA expression was downregulated in the NSCLC tissues compared with the normal lung tissues (*p* < 0.001). A similar trend was observed in the GEO cohort, including GES120622, GSE116959, and GSE19188, as well (Figure 1G–I). Meanwhile, we also investigated the expression of ITGB2 at various stages of tumor progression using the UALCAN database. According to the results, a higher expression level of ITGB2 was observed in tumors at relatively earlier stages (Figure 1C–F). In addition, the Kaplan–Meier analysis revealed that patients with a low ITGB2 expression had a shorter overall survival (OS) time (*p* = 0.01) than those with a high ITGB2 expression (Figure 1J). Overall, these findings suggested that ITGB2 may be a tumor suppressor in NSCLC malignant phenotypes.

### 3.2. ITGB2 Expression Was Downregulated in NSCLC Cell Lines and Tissues

Next, we validated the results of the public database analysis by investigating the expression profile of ITGB2 in the NSCLC cell lines and tissues. We first applied an algorithm to compare the expressions of the ITGB2 gene in the various NSCLC cell lines within the CCLE dataset. A variety of cell lines showed lower levels of ITGB2 expression (Figure 2A). Among them, H1299, A549, PC9, H1975, H1792, DV90, and SK-MES-1 were selected for verification. The results showed that the ITGB2 expression was commonly downregulated in the NSCLC cell lines compared to the lung normal cell line in terms of both the mRNA level and protein level (Figure 2B,C). Moreover, we further confirmed the results for the NSCLC tissues (N = 29 pairs). A QRT-PCR assay was performed to detect the mRNA expression of ITGB2, revealing the significant downregulation of ITGB2 in the NSCLC tissues compared to the adjacent lung tissues (*p* = 0.0013) (Figure 2D). Furthermore, the immunohistochemistry assay showed that the protein level of ITGB2 was lower in the NSCLC tissues compared to the adjacent lung tissues (Figure 2E). Collectively, these results demonstrated that ITGB2 was downregulated in the NSCLC cell lines and tissues, also indicating that the results based on the analysis of the public databases were reliable.

### 3.3. Identification of DEGs

To explore the potential molecular function of ITGB2 in the development of NSCLC, we then identified and compared the differentially expressed genes (DEGs) between the high-expression ITGB2 group and the low-expression ITGB2 group using the TCGA data. A total of 1089 DEGs were identified, with the threshold values of the adjusted *p*-value (p.adj) of <0.05 and |log2 FC| of >2, of which 418 were upregulated and 671 were downregulated, which were presented in volcano plots (Figure 3A). Then, we performed GO functional enrichment and KEGG pathway analyses to determine the biological features of the DEGs. For the upregulated DEGs, the biological process (BP) terms are most significantly enriched in the regulation of leukocyte activation, positive regulation of the immune response, inflammatory response, and cell activation. Additionally, for the downregulated DEGs, the BP terms are most significantly enriched in epithelial cell differentiation, the cellular response to fatty acid, regulation of secretion by cells, and the neuropeptide signaling pathway (Figure 3B). The KEGG enrichment analysis (Figure 3B) revealed that immune-related pathways, together with some metabolism processes, were among the top enriched pathways, including the cytokine–cytokine receptor interaction, cell adhesion molecules, antigen processing and presentation, neutrophil extracellular trap formation, folate biosynthesis, steroid hormone biosynthesis, etc. These biological processes and signaling pathways likely play important roles in the development of NSCLC.

### 3.4. The Aberrant Expression of ITGB2 in NSCLC Is Associated with Immune Cell Infiltration and EMT

To investigate the specific immune contexture in different ITGB2 mRNA expression statuses, we further explored the association between the expression level of ITGB2 and immune cell infiltration level quantified by ssGSEA in NSCLC using Spearman’s correlation. The results showed that ITGB2 expression was positively correlated with the infiltration levels of effector memory CD8 T cells, regulatory T cells, T follicular helper cells, monocytes, MDSC, and natural killer cells (Figure 3C), of which the top two immune cells were Treg cells (rPearson = 0.71, *p* < 0.001) and MDSC (rPearson = 0.73, *p* < 0.001) Figure 3D,E, demonstrates that ITGB2 may be valuable in immunotherapy.

Additionally, an accumulating number of studies have highlighted that EMT plays important roles in tumor progression, metastasis, and drug resistance. We then explored whether the aberrant expression of ITGB2 affected the EMT-associated genes. The Venn diagram analysis revealed that 15 EMT-associated genes were common to the EMT-related gene sets and the DEGs, including ADIPOQ, PIK3CG, CCR2, etc. (Figure 3F). Next, we constructed the PPI network using the DEGs and 15 EMT-associated genes to identify the hub genes of 15 EMT-associated genes. The results revealed that 10 of 15 EMT-associated genes can be considered as the hub genes, 7 as upregulated (ADIPOQ, PIK3CG, CCR2, CCR7, CSF2, LIN28A, and MMP9) and 3 as downregulated (TLR4, IRS2, and CD14), through which ITGB2 likely affected the NSCLC metastasis (Figure 3G), illustrating the potential correlation between ITGB2 and EMT.

### 3.5. ITGB2 Inhibits NSCLC Cell Proliferation, Migration, and Invasion In Vitro

To elucidate the effect of ITGB2 on NSCLC cell proliferation in vitro, H1792 cells were infected with LV-ITGB2. As expected, both ITGB2’s mRNA transcription and protein expression were significantly increased in the H1792 cells (Figure 4A). A CCK-8 assay was used to measure the cell viability and found that the overexpression of ITGB2 dramatically inhibited the H1792 cell proliferation (Figure 4B). We also evaluated the effect of ITGB2 on the proliferation of NSCLC cells by an EdU assay, and the result was consistent with the foregoing result of the CCK-8 assay (Figure 4C). Next, we examined the effect of ITGB2 on lung cancer cell migration and invasion using Transwell assays. The result showed that the overexpression of ITGB2 significantly suppressed the migration and invasion of the H1792 cells (Figure 4D). The Western blot analysis suggested that ITGB2 overexpression decreased the expressions of N-cadherin, Vimentin, Slug, Twist, and Snail and increased the expression of E-cadherin (Figure 4E). As a result of these findings, ITGB2 protein expression appears to be negatively correlated with EMT.

## 4. Discussion

NSCLC is the most common type of lung cancer, representing about 85% of lung cancer cases [26]. In recent years, although a number of studies have been conducted to explore the treatment and mechanism of NSCLC, the overall survival of patients is still less than 5 years [27]. Thus, it is particularly important to find new molecular markers for the prognosis of lung cancer patients. In this study, we proved that the expression of ITGB2 in NSCLC tissue was significantly reduced compared with adjacent lung tissues and was related to the stage of NSCLC patients. In addition, the low expression of ITGB2 was associated with worse overall survival. All these results suggested that ITGB2 may be a molecular diagnostic target or therapeutic marker for LUAD and LUSC patients.

According to previous studies, ITGB2 is related to the occurrence of many diseases, including carotid atherosclerosis, necrotizing enterocolitis, and cancer. The expression of ITGB2 was upregulated in atheroma plaque compared with intact tissue and involved in the development of carotid atherosclerosis [28]. Moreover, ITGB2 deficiency causes the hyperresponsiveness of the aberrant Toll-like receptor and leads to necrotizing enterocolitis pathogenesis [29]. In cancer studies, ITGB2 was found to be significantly elevated in high-malignant gliomas and related to a worse prognosis [20]. In addition, ITGB2 was proved to promote the effects of the migration capability and invasion capability of breast cancer via biological function experiments [30]. ITGB2 was identified to be the key immune-related gene in acute myeloid leukemia patients, while the high expression of ITGB2 was related to the poor prognosis of acute myeloid leukemia patients [31]. These results indicated that ITGB2 plays significant roles in tumorigenesis and progression. However, there is no research on the biological function and association of ITGB2 expression in NSCLC at present. In our research, we explored the expression of ITGB2 in tumor and normal tissues, and the results confirmed that a higher expression of ITGB2 was correlated with a better prognosis in patients, suggesting that ITGB2 may be used as a molecular diagnostic marker in LUAD and LUSC. The immunohistochemistry assay showed that ITGB2 was expressed higher in normal tissues than in cancer tissues. The biological function analysis showed that the overexpression of ITGB2 can inhibit the proliferation, migration, and invasion ability of tumor cells.

The composition of the tumor-infiltrating cell population plays a critical role in tumor progression and the immunotherapeutic response [32]. Immune regulation is recognized as being related to the prognosis of patients [33]. We analyzed the correlation between ITGB2 and various tumor-infiltrating cells and found that ITGB2 was positively correlated with MDSC and regulatory T cells. MDSC is the heterogeneous population of immature myeloid cells. High levels of MDSC promote tumor immune escape from the tumor microenvironment by reducing the T cells’ activity. MDSC protects the tumor cells from the patient’s immune system, leading to resistance to immunotherapy [34,35]. Regulatory T cells suppress the aberrant immune response against self-antigens and anti-tumor immune response. An increased infiltration level of regulatory T cells is associated with a poor prognosis of the patients [36]. The difference in the infiltration of the microenvironment may contribute to the poorer prognosis and cancer progression of patients. To further explore the potential mechanisms and possible molecular function of ITGB2 in lung cancer tumorigenesis, we performed GO and KEGG analyses to explore the pathways enriched in the differentially expressed genes of ITGB2-DEGs. THe KEGG and GO enrichment results showed that the DEGs-up were mainly enriched in the hematopoietic cell lineage, cell adhesion molecules, and regulation of leukocyte activation, and the DEGs-down were mainly enriched in the neuroactive ligand–receptor interaction and epithelial cell differentiation. A number of studies have shown that adhesion-related molecules are involved in the regulation of the epithelial–mesenchymal transition process and promotion of tumor invasion and metastasis [37]. Then, 1089 DEGs of ITGB2 and 305 EMT-related genes were obtained for the analysis. Finally, 15 key genes were screened, and a PPI network was constructed for further study. Metastasis is one of the major causes of cancer deaths. EMT is implicated in tumorigenesis and the improved metastatic properties of cancer cells by enhancing their mobility and invasion [38]. Through in vitro experimental studies, we confirmed that the overexpression of ITGB2 can increase the expression level of E-cadherin, while it decreases the expression levels of N-cadherin, Vimentin, Slug, Snail, and Twist in H1792 cells, indicating that ITGB2 can affect the growth of NSCLC cells through inhibiting the EMT pathway.

## 5. Conclusions

We used a series of bioinformatics analyses, the GEO and TCGA databases, and our cohort to show that the expression of ITGB2 was downregulated in patients with NSCLC. The GO and KEGG enriched analyses showed that the expression of ITGB2 was associated with immune-related pathways and some metabolic processes. The immune cell infiltration analysis revealed that the expression of ITGB2 was positively correlated with the infiltration of Treg cells and MDSC in NSCLC. In the cellular models, ITGB2 overexpression inhibited the proliferation, migration, and invasion of the NSCLC cell lines. The overexpression of ITGB2 increased the expressions of N-cadherin, Vimentin, Slug, Twist, and Snail and decreased the expression of E-cadherin, thereby possibly inhibiting NSCLC cell proliferation, migration, and invasion (Figure 4F). Therefore, increasing the expression of ITGB2 in vivo could find application as a novel strategy in the treatment of NSCLC.

## Figures and Tables

**Figure 1 jcm-11-06421-f001:**
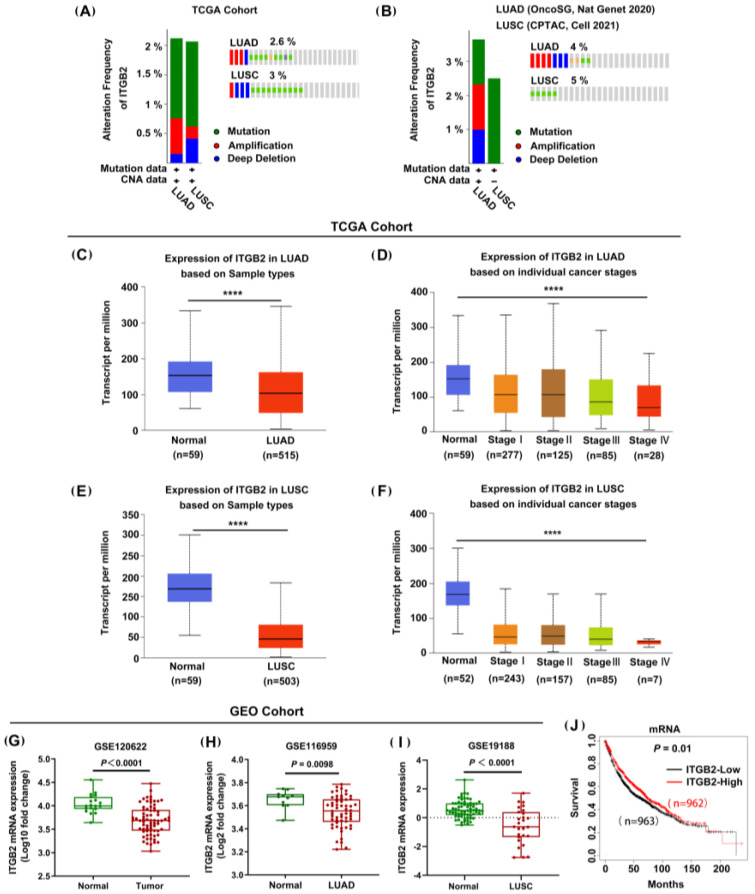
ITGB2 expression was downregulated in NSCLC, and its low levels were associated with a poor prognosis in the public databases. (**A**,**B**) Genetic changes in ITGB2, analyzed by cBioPortal, based on the TCGA lung cancer dataset (**A**) and OncoSG and CPTAC datasets (**B**). (**C**,**D**) Expression of ITGB2 in LUAD tissue and adjacent normal tissue (**C**) and LUAD of different tumor stages (**D**) in the TCGA database (UCLCAN). (**E**,**F**) Expression of ITGB2 in LUSC tissue and adjacent normal tissue (**E**) and LUSC of different tumor stages (**F**) in the TCGA database (UCLCAN). (**G**–**I**) Expression of ITGB2 in NSCLC tissue and adjacent normal tissue (**G**), in LUAD tissue and adjacent normal tissue (**H**), and in LUSC tissue and adjacent normal tissue (**I**) according to the GEO cohort. (**J**) Kaplan−Meier survival curves for the overall survival (OS) according to ITGB2’s expression levels in NSCLC. **** *P* < 0.0001.

**Figure 2 jcm-11-06421-f002:**
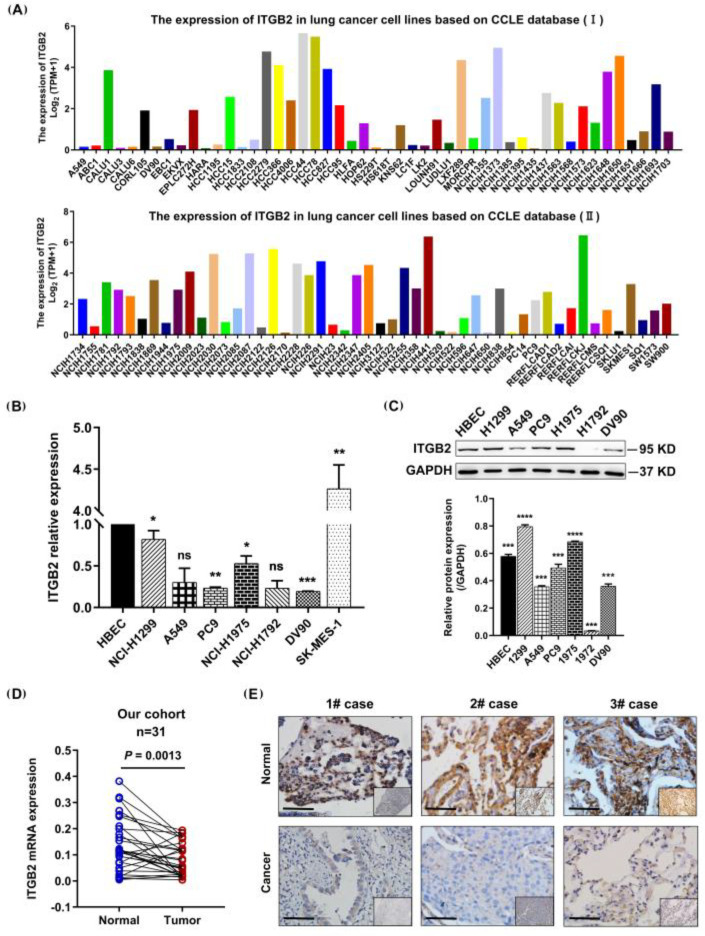
ITGB2 was commonly downregulated in the NSCLC cell lines and tissues. (**A**) Expression of ITGB2 in NSCLC cell lines in CCLE database. (**B**,**C**) ITGB2 expression at mRNA level (**B**) and protein level (**C**) in NSCLC cell lines. (**D**) The pairing difference analysis of ITGB2 expression between tumor and normal tissues (*n* = 31). (**E**) Immunohistochemical staining of ITGB2 in tumor and non-tumor tissues from patients with NSCLC. Scale bars are shown in the figure (scale bars, 20 μm). * *P* < 0.05, ** *P* < 0.01, *** *P* < 0.001, **** *P* < 0.0001.

**Figure 3 jcm-11-06421-f003:**
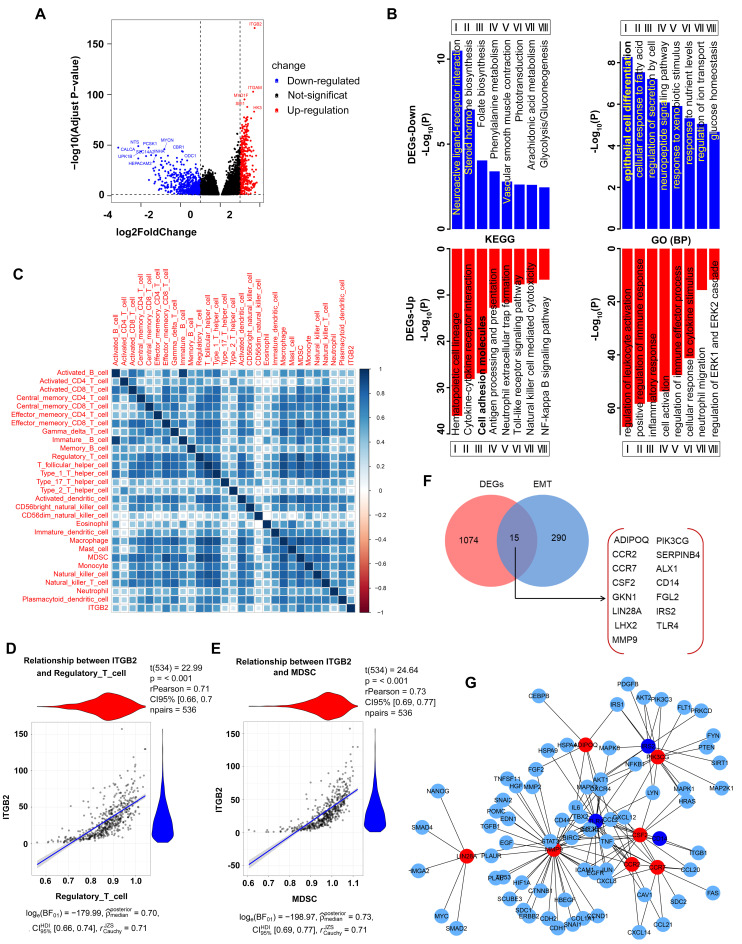
The aberrant expression of ITGB2 in NSCLC is associated with immune cell infiltration and EMT. (**A**) Four hundred and eighteen upregulated genes and six hundred and seventy−one downregulated genes were identified in the high-expression ITGB2 group based on the TCGA database. (**B**) GO (BP) and KEGG enrichment annotation of the DEGs in the high−expression ITGB2 group vs. low−expression ITGB2 group. (**C**) Analysis of the correlations between the aberrant expression of ITGB2 and immune cell infiltration in NSCLC. (**D**,**E**) The positive correlation between ITGB2 and Treg cells (**D**) and MDSC (**E**). (**F**) Venn diagram analysis of EMT−related genes in DEGs. (**G**) Protein−protein interaction (PPI) network constructed with the DEGs. Red nodes represent upregulated EMT−related genes, and blue nodes represent downregulated EMT−related genes.

**Figure 4 jcm-11-06421-f004:**
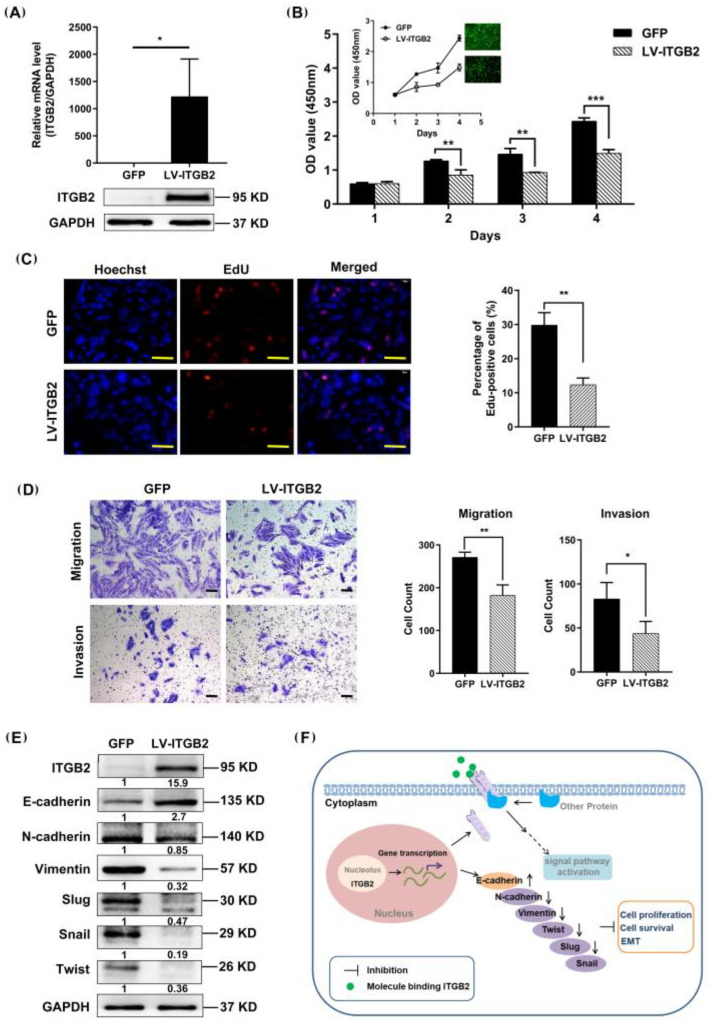
ITGB2 inhibits the proliferation, migration, and invasion of NSCLC cells in vitro. (**A**) ITGB2 expression at the mRNA level and protein level in H1792 cells. (**B**) Cell viability was detected by CCK-8 assay (** *p* < 0.01, *** *p* < 0.001), and the optical density (OD) absorbance values at 450 nm were measured using a microplate reader, embodying the proliferation capacity of the cells. (**C**) Cell proliferation ability was detected by the EdU assay, scale bars: 50 μm (** *p* < 0.01). (**D**) The migration and invasion of H1792 cells were evaluated by a Transwell assay, scale bars: 100 μm, (* *p* < 0.05, ** *p* < 0.01). (**E**) The Epithelial–mesenchymal transition markers were detected by Western blot assay in H1792 cells. (**F**) Schematic diagram of the potential mechanisms of ITGB2-mediated inhibition of tumorigenesis in NSCLC. All representative images from at least three independent experiments. * *P* < 0.05, ** *P* < 0.01, *** *P* < 0.001.

## Data Availability

The datasets used and analyzed during this study are available from the Cancer Cell Line Encyclopedia (https://sites.broadinstitute.org/ccle/datasets, accessed on 1 May 2022), The Cancer Genome Atlas (TCGA) (https://www.cancer.gov/, accessed on 1 May 2022), and Gene Expression Omnibus (GEO) (https://www.ncbi.nlm.nih.gov/geo/, accessed on 1 May 2022).

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
