# Peer review of "The Profile and Clinical Significance of ITGB2 Expression in Non-Small-Cell Lung Cancer"

_jcm, 2022, doi:10.3390/jcm11216421_

Round 1

Reviewer 1 Report (Previous Reviewer 2)

The manuscript has been revised well.

Reviewer 2 Report (Previous Reviewer 1)

I thank the authors for the changes. The paper is strongly improved.

This manuscript is a resubmission of an earlier submission. The following is a list of the peer review reports and author responses from that submission.

Round 1

Reviewer 1 Report

This is a very interesting study regarding the TME expressions on patients with NSCLC. I think that this is a well exposed manuscript with an novel analysis that can aid to define now prognostic factors for NSCLC. I have only a question for the author. The public databases from which the authors have done the analysis are composed of early stage and advanced stage patients. Which kind of patient the authors included? Please specify it in the methods.

Author Response

Thanks for the suggestion. In this work, we collected both early and advanced stage of NSCLC patients for analysis. We have revised it as shown in “Materials and Methods-2.1 Data Collection and Processing” marked by red font. Please see the the revised manuscript in attachment.

Reviewer 2 Report

In this paper, the authors examine the significant of ITGB2 expression in non-small cell lung cancer. I read the study with interest. The data are novel and informative.

In addition, the manuscript is well written and this is a potentially interesting manuscript.

I do not have any substantial amendments to suggest. I have a few relatively comments, explained below.

Minor

1.      The letters of Fig. 2a and the graph of Fig. 2b are extremely difficult to evaluate as it is hardly visible.

2.      How many times was the experiment repeated?

3.      In Figure 4b, what does “OD value” mean? Please add the explanation.

Author Response

1. According to your suggestion, we have adapted the whole layout of Fig.2 and re-uploaded Fig.2 with high quality, making it easy for readers to evaluate the results. Please view the Fig.2 in revised manuscript.

2. Every experiment had been repeated three times to conduct calculation. And we have indicated in methods and legend, which was marked by red font.

3. “OD value (450nm)” means the optical density of Formazan, embodying the proliferation ability of the detected cells. And we have added the explanation in legend of Fig.4 and marked it by red font.

Please see the the revised manuscript in attachment.

Reviewer 3 Report

Integrins are heterodimeric receptors composed of alpha and beta chains, for many integrins there is an intracellular pool of beta subunit, and it is the alpha subunit that determines expression levels, i.e. levels of beta subunit is often in excess.

Beta 2 integrins are found on cells of the immune system, not in other cell types. Cultured cells might express RNA of b2 integrins and low levels of corresponding protein, but this is in vitro artefact. Reports of role of b2 integrins in tumor cells are published in low impact journals and have not been confirmed by established groups specializing in integrin biology. When expressed in immune cells b2 integrins need to be activated via inside-out signaling.

The paper of Zu et al now claims that b2 integrin play a role in lung cancer cells.

In summary:

1. The study reveals lack of knowledge in integrin biology.

2. Given that b2 integrins are specific to immune cells the staining in Figure 2 is not convincing. Staining of a non-inflamed tissue vs inflamed tissue with leukocytes should be used as control. Published leukocyte cell lines with high level expression of b2 integrins should have served as control for cell detection levels. Alternatively, knock-down of ITGB2 should be used to confirm specificity of staining.

3. The authors should have established which alpha chain is associated with the b2 chain. Integrins exist as functional heterodimers at the cell surface.

4. In 3.3 it is established in DEG analysis that a subunit of an immune cell receptor (ITGB2) stimulates immune-related pathways. Given the reported leukocyte-specific expression of ITGB2, this is maybe not surprising. This is in fact what you expect of a leukocyte receptor.

5. ITGB2 is synthesized as a precursor, it is unclear which band is shown in the Western in Figure 4, precursor band or the mature band? By immunoprecipitating with an integrin alpha chain antibody this issue can be resolved.

Author Response

Point 1: The study reveals lack of knowledge in integrin biology.

Response 1: Thanks for the suggestions. We will further accumulate and enrich the greater knowledge in integrin biology for ourselves and we have made some supplement in introduction which is marked by red font.

Point 2: Given that b2 integrins are specific to immune cells the staining in Figure 2 is not convincing. Staining of a non-inflamed tissue vs inflamed tissue with leukocytes should be used as control. Published leukocyte cell lines with high level expression of b2 integrins should have served as control for cell detection levels. Alternatively, knock-down of ITGB2 should be used to confirm specificity of staining.

Response 2: Thank you for your suggestions. Although b2 integrin is commonly regarded to be specific to immune cells, an increasing number of studies have reported that ITGB2 has been detected in multiple malignant solid tumor cells and stromal cells including breast cancer cells (MCF10A, MDA-MB-231)[1], nasopharyngeal carcinoma (CNE1, 6-10B)[2], as well as CAF cells[3]. In fact, ITGB2 is not the first leukocyte-specific expressing gene detected in multiple malignant solid tumor cell lines. For instance, Aiolos (encoded by IKZF3) is a member of the Ikaros zinc finger family, the expression of which is normally restricted to lymphoid cells. However, it had been reported to express in multiple malignant solid tumor cell lines including MCF-7, SW480, HEK, PC3, and HeLa after then[4]. Moreover, in our work, we examined the expression of ITGB2 was downregulated in cancer tissues compared to normal tissues based on the public and our own data by multiple validations. Eliminating the stereotype , our results, especially the staining of ITGB2 in tissues and the band shown in the Western blot, are credible.

Point 3: The authors should have established which alpha chain is associated with the b2 chain. Integrins exist as functional heterodimers at the cell surface.

Response 3: Thank you for your suggestions. In this work, we aim to explore the expression profile and the role of ITGB2 in NSCLC. We performed a series of experiments including bioinformatic analysis based on different datasets, qRT-PCR and IHC assay based on lung cancer tissues, as well as the biological functional assay based on cell level, to elucidate the expression profile and the role of ITGB2 in NSCLC. In this study, we did not explore that which alpha chain is associated with the b2 chain, but it will be a good scientific question and another study topic of our further in-depth research.

Point 4: In 3.3 it is established in DEG analysis that a subunit of an immune cell receptor (ITGB2) stimulates immune-related pathways. Given the reported leukocyte-specific expression of ITGB2, this is maybe not surprising. This is in fact what you expect of a leukocyte receptor.

Response 4: Thank you for your suggestions. We think that the experimental results should be recorded and descripted objectively. This result may be not surprising, but it’s true, indeed. Given the reported leukocyte-specific expression of ITGB2, it means that ITGB2 plays important roles in immune-related pathways. Taking it into account, the DEG analysis exhibited a subunit of an immune cell receptor (ITGB2) stimulated immune-related pathways, reflecting that the results of DEG analysis are credible. In addition to the immune-related pathways might be influenced by leukocyte, the remaining            enrichment may indicated that glycolysis or folate biosynthesis regulated tumorigenesis of NSCLC affected by tomor cell itself.

Point 5: ITGB2 is synthesized as a precursor, it is unclear which band is shown in the Western in Figure 4, precursor band or the mature band? By immunoprecipitating with an integrin alpha chain antibody this issue can be resolved.

Response 5: ITGB2 protein is composed of 769 aa and the molecular weight is 95 KD. It had been proved that the mature β subunit weighted 95 KD, while the precursor of it varied.[5-7] In accordance with the reliable researches, the ITGB2 in our western shown in Figure 4 is the mature band.

References

  1. Liu, H.; Dai, X.; Cao, X.; Yan, H.; Ji, X.; Zhang, H.; Shen, S.; Si, Y.; Zhang, H.; Chen, J.; et al. PRDM4 mediates YAP-induced cell invasion by activating leukocyte-specific integrin beta2 expression. EMBO reports 2018, 19, doi:10.15252/embr.201745180.
  2. Li, J.; Zhang, Z.; Feng, X.; Shen, Z.; Sun, J.; Zhang, X.; Bu, F.; Xu, M.; Tan, C.; Wang, Z. Stanniocalcin-2 promotes cell EMT and glycolysis via activating ITGB2/FAK/SOX6 signaling pathway in nasopharyngeal carcinoma. Cell biology and toxicology 2022, 38, 259-272, doi:10.1007/s10565-021-09600-5.
  3. Zhang, X.; Dong, Y.; Zhao, M.; Ding, L.; Yang, X.; Jing, Y.; Song, Y.; Chen, S.; Hu, Q.; Ni, Y. ITGB2-mediated metabolic switch in CAFs promotes OSCC proliferation by oxidation of NADH in mitochondrial oxidative phosphorylation system. Theranostics 2020, 10, 12044-12059, doi:10.7150/thno.47901.
  4. Billot, K.; Parizot, C.; Arrouss, I.; Mazier, D.; Debre, P.; Rogner, U.C.; Rebollo, A. Differential aiolos expression in human hematopoietic subpopulations. Leukemia research 2010, 34, 289-293, doi:10.1016/j.leukres.2009.05.016.
  5. F. Sanchez-Madrid, J.A. Nagy, E. Robbins, P. Simon, T.A. Springer, A human leukocyte differentiation antigen family with distinct alpha-subunits and a common beta-subunit: the lymphocyte function-associated antigen (LFA-1), the C3bi complement receptor (OKM1/Mac-1), and the p150,95 molecule., J. Exp. Med. 158 (6) (1983) 1785-1803. https://doi.org/ 10.1084/jem.158.6.1785.
  6. F. Sanchez-Madrid, A.M. Krensky, C.F. Ware, E. Robbins, J.L. Strominger, S.J. Burakoff, T.A. Springer, Three distinct antigens associated with human T-lymphocyte-mediated cytolysis: LFA-1, LFA-2, and LFA-3, Proc Natl Acad Sci U S A 79 (23) (1982) 7489-7493. https://doi.org/ 10.1073/pnas.79.23.7489.
  7. T.K. Kishimoto, N. Hollander, T.M. Roberts, D.C. Anderson, T.A. Springer, Heterogeneous mutations in the β subunit common to the LFA-1, Mac-1, and p150,95 glycoproteins cause leukocyte adhesion deficiency, Cell 50 (2) (1987) 193-202. https://doi.org/ 10.1016/0092-8674(87)90215-7.

Round 2

Reviewer 3 Report

I rejected and not accepted all the authors' responses to my comments.